## Article

# Uncovering the Drivers and Regional Variability of Cotton Yield in China

Yaqiu Zhu [1], Bangyou Zheng [2], Qiyou Luo [1], Weihua Jiao [3] and Yadong Yang [1,*]

1 State Key Laboratory of Efficient Utilization of Arid and Semi-Arid Arable Land in Northern China, Institute of Agricultural Resources and Regional Planning, Chinese Academy of Agricultural Sciences, Beijing 100081, China; zhuyaqiu@foxmail.com (Y.Z.); luoqiyou@caas.cn (Q.L.)
2 CSIRO Agriculture and Food, Queensland Bioscience Precinct, 306 Carmody Rd., St. Lucia, Queensland 4067, Australia; bangyou.zheng@csiro.au
3 School of Economics, Shandong University of Finance and Economics, Jinan 250014, China; caujwh@163.com
* Correspondence: yangyadong@caas.cn

**Abstract:** Cotton (*Gossypium hirsutum* L.) is an economically important crop in China, and responses of cotton yield in different regions to separate and joint changes in natural and anthropogenic factors are the foundation for sustainable development under climate change; however, these remain uncertain. Here, we analyzed the spatiotemporal evolution and heterogeneity of cotton cultivation in China from 1949 to 2020 and quantified the response of cotton yield variations in air temperature, precipitation, solar radiation, disaster, and crop management factors between 1980 and 2020 by the Pettitt mutation test and GeoDetector. Multi-site meteorological data were obtained from different cotton-growing regions and corresponding cotton yield and phenology data were obtained from provinces. Our findings showed that all 17 Chinese provinces experienced advancements in cotton yield. Relative to 1949–1967, China's cotton production in 2007–2020 increased by 400% while cotton yield increased by 420%. Increases in factors such as minimum temperature (TES), average temperature (ADT), effective accumulated temperature (EAT), precipitation (PP), daily solar radiation (SSD), non-farm employment opportunities (O), disaster area (D), geographic region (GEO) and agricultural technologies like fertilizer usage (F), genetically modified varieties (Bt), and mechanized farming (M) have contributed to the enhanced cotton yield. The importance of single factors influencing cotton yield of China in descending order was as follows: F > Bt > M > GEO > EAT > O > PP > TES > ADT > SSD > D. However, the effects of different climatic and agriculture technological elements on cotton yield are spatially heterogeneous by region, and the combined effects of those elements are higher than those of single elements. The effects of driving factors vary across regional scales. The most significant interaction effects were observed between chemical fertilizer use and other driving factors. Specifically, the interaction between F and TES has the greatest explanatory influence in Northwest China. Our findings provide a reference for the development of more accurate adaptation strategies and management measures in different regions. We recommend that policymakers prioritize measures such as improving climate-resilient cotton varieties, encouraging technological advancements, and implementing policies that support equitable distribution of cultivation.

**Keywords:** climate change; cotton phenology; technological advancement; minimum temperature; GeoDetector





## 1. Introduction

Cotton is a vital economic crop for many developing countries. Cotton connects the two fields of agriculture and textiles and is a source of livelihood for millions of smallholders, workers, and their families. The cotton business contributes significantly to the economies of many developing countries. In 1986, the top three cotton-consuming countries globally were China, the Soviet Union, and India, with cotton consumption accounting for 23.7%, 11.4%, and 9.6% of the global total, respectively. By 2016, China,

India, and Pakistan had cotton consumption rates of 33.1%, 20.9%, and 9.1% [1]. In 2021, China's cotton production, imports, and exports accounted for about 22%, 2.5%, and less than 1% of the world's total, respectively [2]. China has remained the world's largest cotton-consuming nation since 1986.

Genetic improvements in cotton varieties contributed strongly to increases in cotton yield in China [3]. Bacillus thuringiensis (Bt) cotton, which is the most successful commercialization of a GM crop, was first planted in China in 1997 [4]. By 2014, the planting area of Bt cotton was 3.9 million hectares, which was 93% of the total cotton production in China, and was planted by 7.1 million small-scale farmers [5]. Moreover, studies show that India's genetically modified (GM) crop produces higher yields and has expanded agricultural land in certain circumstances. In 2019, there was a renewed increase in the average adoption rate of biotechnology crops, with India leading at a remarkable 95% acceptance rate. Moreover, the reduced reliance on pesticides in GM crop cultivation holds the potential to reduce the emission of gases into the environment [6,7]. Over the past few decades, China has undergone rapid changes in land use and significant shifts in crop management approaches [8]. Natural resources and anthropogenic activities are fundamentally altering the global spatial distribution of crop cultivation [9]. Crop growth is influenced by various factors, including global warming [10,11], soil nutrients [12,13], and the development and utilization of water resources [14]. Simultaneously, cultivation management practices are shaped by combinations of natural- and anthropogenic-induced elements [15–17].

China's cotton cultivation currently faces severe challenges of water and soil resource constraints and market competition. Due to factors such as climate change, urbanization, an aging rural population, increasing production costs, and declining comparative efficiency [3,18–20], 85% of the former cotton area has been converted to other uses, and its production centers have rapidly shifted to new farmland in Xinjiang [21]. Amid challenges like water and soil resource constraints and climate change, achieving high and stable cotton production in China is a pressing concern. It is thus crucial to uncover the primary drivers of regional variability in anthropogenic–natural impacts on cotton to achieve high and stable yields in China.

This study explores how various factors, such as climate, agricultural practices, and disaster elements, interact and influence cotton yield across diverse geographic regions. The objectives of this study are to (i) characterize the trends in China's cotton cultivation areas and yields between 1949 and 2020; (ii) identify shifts in the temporal–spatial variations in cotton yields in 17 Chinese provinces during different time periods; and (iii) examine the influences of single human–natural factors and their interactions on variations in cotton yield across various regional scales between 1980 and 2020.

## 2. Materials and Methods

### 2.1. Research Data and Region

The data used in this study mainly included cotton data, agriculture technical data, climatic data, disaster data, spatial data, and cotton phenology data. Data sources include statistical yearbooks, statistical websites, and the literature. Cotton data include cotton area, yield, and production in mainly cotton planting regions from 1949 to 2020, which were from China Cotton Statistics Compilation, China Rural Statistical Yearbook, and China Statistical Yearbook [https://data.cnki.net/] on 1 August 2022. The cotton phenology data, which record different stages of cotton occurrence, were derived from the dataset for crop growth and development status in China, provided by the National Meteorological Information Center of the China Meteorological Administration [https://cdc.cma.gov.cn/] on 1 August 2021. This dataset includes information from 64 agricultural meteorological observation stations located in the main cotton-planting regions. Climatic data were obtained from the National Meteorological Information Center [https://cdc.cma.gov.cn/] on 1 August 2021 and encompassed daily averages, maximum and minimum air temperatures, solar radiation, and precipitation recorded at 389 meteorological observation stations between 1980 and 2020 (Figure 1).

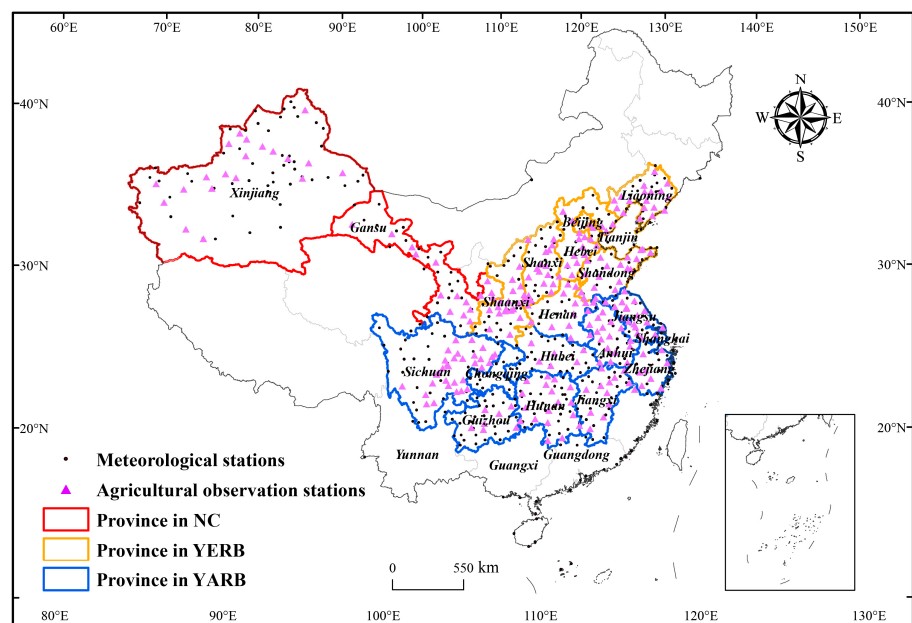

**Figure 1.** Study area, showing the location of meteorological stations (dots) and agricultural meteorological observation stations of cotton cultivation (triangles) in the study area. There are 139, 185, and 65 meteorological stations in YERB, YARB, and NC, respectively. There are 64 agricultural meteorological stations for cotton cultivation in China. The base map was applied without endorsement using data from the National Geomatics Center of China [NGCC; http://www.ngcc.cn/ngcc/] and the Institute of Agricultural Resources and Regional Planning, Chinese Academy of Agricultural Sciences [IARRP; https://iarrp.caas.cn/] on 1 August 2022.

Disaster data, including droughts and floods that led to reductions of over 10% in cotton production area, primarily originated from the China Statistical Yearbook [https://data.cnki.net/] on 1 August 2022. Agricultural technical data, encompassing information on chemical fertilizer usage, total power of agricultural machinery, and Bt transgenic insect-resistant cotton, spanning from 1980 to 2020, were mainly sourced from the China Statistical Yearbook and the National Compilation of Information on the Costs and Benefits of Agricultural Products [https://data.cnki.net/] on 1 October 2022, as well as the Ministry of Agriculture and Rural Affairs of the People's Republic of China [http://www.gov.cn/fuwu/bm/nyncb/index.htm] on 1 October 2022.

The spatial data are divided into three regions, Northwest China (NC, 32–50° N, 75–108° E), the Yellow River Basin (YERB, 31–43° N, 105–125° E), and the Yangtze River Basin (YARB, 24–35° N, 97–122° E), from 1980 to 2020. These data allow us to analyze how human–natural factors affect the spatial pattern of cotton cultivation. The three regions accounted for more than 99% of cotton production on a national scale over this time period [18] and have diverse climatic conditions [22]. NC and YERB share a middle and warm temperate temperature zone, while YARB features a subtropical climate. In terms of wet/dry climatic regions, NC falls under the arid category, YERB experiences a semi-arid climate, and YARB enjoys a humid climate. Sunshine duration varies across these regions, with NC receiving 2600 to 3400 h of annual sunshine, YERB ranging from 1900 to 3000 h, and YARB having 1200 to 2500 h. Precipitation patterns show diversity, with NC receiving 15–380 mm annually, YERB receiving 400–1000 mm, and YARB experiencing 1000–1600 mm. The accumulation of temperature above 10 °C indicates 3000 to 5399 °C·d for NC, 2600 to 4899 °C·d for YERB, and 4600 to 5999 °C·d for YARB. Finally, annual temperature ranges for NC, YERB, and YARB are 30.8–43.4 °C, 21.3–39.8 °C, and 21.4–26.6 °C, respectively.

### 2.2. Conceptual Framework: Determinants of Cotton Yield

Crop production at the field level is commonly represented as a technical relationship between yield and yield-affecting factors, including climate, input utilization, soil quality, and agricultural management practices. While climate and soil fertility are determined externally for crop growth, farmers have control over input usage and agricultural management practices [23]. Cotton growth are under the comprehensive influence of internal and external conditions, including the effects of other changes [24].

The natural factors included geographic region (GEO), average daily temperature (ADT), sum of the daily precipitation (PP), sum of the daily solar radiation (SSD), sum of the daily temperature between 12 °C and 36 °C (EAT), average of the minimum daily temperature (TES/TFB), average of the maximum daily temperature (MAT), and the proportion of natural disasters (D), which related to geographic location, climate, and disasters, determining the stability and fluctuation of cotton production directly [25,26]. Rising temperatures have a significant impact on cotton seed germination rates. An increase in daytime maximum temperatures accelerates cotton photosynthesis, leading to increased yields [18,19]. Furthermore, higher daytime minimum temperatures expand the suitable climate range for cotton cultivation in the region [27]. Precipitation also affects cotton, influencing both seed rot rates and boll counts [28,29].

Human factors related to technological advancements indirectly influence agricultural production by altering land configuration, agricultural machinery, and genetic structures. These changes subsequently impact cotton cultivation processes and yield. Technological advancements in cotton cultivation encompass three primary indicators: chemical fertilizer (F) usage, total power of agricultural machinery (M), and the adoption of Bt transgenic insect-resistant cotton (Bt).

Increased application of chemical fertilizers correlates with higher cotton yields, making the cost of chemical fertilizer per unit area a useful gauge of fertilizer development levels. The total power of agricultural machinery indicates the level of mechanization in cotton production. The utilization of genetically modified Bt cotton enhances yield; considerable evidence suggests that Bt cotton has brought economic benefits to farmers in numerous countries [30]. Starting from the successful development of domestically produced anti-insect cotton (GK) in China in 1994, which effectively eradicates cotton bollworms, the commercial cultivation of insect-resistant cotton was approved by the Ministry of Agriculture in 1997. Pilot programs were initiated in Anhui, Shandong, and Shaanxi in 1998, expanding the following year to include provinces such as Jiangsu, Hubei, Hunan, Henan, Jiangxi, and Xinjiang. Over the past two decades, anti-insect cotton has been widely promoted and applied across China. Given the variable approval times for the regional adoption of genetically modified insect-resistant cotton, the year of the first biosafety certificate approval for Bt transgenic insect-resistant cotton in each region was selected as the reference point.

From the perspective of natural and human factors, thirteen driving factors were selected based on previous studies [18,31–33]. Additional information can be found in Table 1.

**Table 1.** Definition of influence factors.

| Type | Variables | Codes | Meaning and Assignment of Variables |
|---|---|---|---|
| Dependent variables | Cotton yield | — | Total cotton production/the area of cotton cultivation (kg/hm$^2$) |
| Spatial indicators | Geographic region | GEO | The cotton-growing geographic regions in which the province is located (categorical variables) |
| Dependent variables | Cotton yield | — | Total cotton production/the area of cotton cultivation (kg/hm$^2$) |
| Spatial indicators | Geographic region | GEO | The cotton-growing geographic regions in which the province is located (categorical variables) |

**Table 1.** *Cont.*

| Type | Variables | Codes | Meaning and Assignment of Variables |
|---|---|---|---|
| Climatic indicators | Average daily temperature from *Sow* to *Squ* | ADT | Average of the daily temperature from *Sow* to *Squ* in the growth period (°C) |
| | Precipitation | PP | Sum of the daily precipitation from *Squ* to *Bol* in the growth period (mm) |
| | Solar radiation | SSD | Sum of the daily sunshine duration from *Sow* to *Mat* in the growth period (hours) |
| | Effective accumulated temperature | EAT | Sum of the daily temperature between 12 °C and 36 °C from *Sow* to *Mat* in the growth period (°C·d) |
| | Minimum temperature from *Eme* to *Squ* | TES | Average of the minimum daily temperature from *Eme* to *Squ* in the growth period (°C) |
| | Minimum temperature from *Flo* to *Bol* | TFB | Average of the minimum daily temperature from *Flo* to *Bol* in the growth period (°C) |
| | Maximum temperature from *Flo* to *Bol* | MAT | Average of the maximum daily temperature from *Flo* to *Bol* in the growth period (°C) |
| Disaster indicators | Disaster area | D | Natural disasters such as drought and floods have reduced cotton production by more than 10% ($hm^2$) |
| Technical indicators | Chemical fertilizer use | F | Cost of chemical fertilizer per mu (mu $yuan^{-1}$) * |
| | Total power of agricultural machinery | M | Total power of agricultural machinery/crop sown area (kWh $hm^{-2}$) |
| | Bt transgenic insect-resistant cotton | Bt | The year of first approval of the biosafety certificate for Bt transgenic insect-resistant cotton in the region; Bt = 1, otherwise, Bt = 0 |
| Social indicators | Non-farm employment opportunities | O | Value-added by secondary and tertiary industry/gross regional product (%) |

* Note: 1 mu = 667 $m^2$.

### 2.3. Methods

2.3.1. Climate Factor Calculations by Cotton Phenology

Cotton cultivation in China involves a series of vital phenological stages, each integral to the cotton plant's growth. These stages encompass sowing (*Sow*), emergence (*Eme*), squaring (*Squ*), flowing (*Flo*), boll opening (*Bol*), and maturity (*Mat*), with precise water and temperature conditions steering their progression [18,26]. In order to dissect the impact of climatic factors on cotton yield variation across distinct regions, this study computes temperature, precipitation, and solar radiation based on cotton phenology.

The primary cotton phenological stages within the major cotton cultivation regions were analyzed through descriptive statistics, and the outcomes are presented in Table S1. Within this study, we focused on three significant cotton-producing areas and defined specific dates for evaluating climatic factors. In Northwest China (NC), sowing typically occurs around 15 April, followed by emergence around 30 April, squaring around 20 June, flowering around 10 July, boll opening by 5 September, and maturity around 20 October. In the Yellow River Basin (YERB), the corresponding dates are 10 April for sowing, 25 April for emergence, 20 June for squaring, 15 July for flowering, 1 September for boll opening, and 30 October for maturity. In the Yangtze River Basin (YARB), sowing starts around 5 April, with emergence occurring by 30 April, squaring by 10 June, flowering by 10 July, boll opening by 1 September, and maturity by 20 October. Notably, these regions showcase diverse geographical and climatic traits in China. Northwest China (NC) witnesses a delayed sowing period, whereas the Yellow River Basin (YERB) and the Yangtze River Basin (YARB) feature earlier sowing dates.

### 2.3.2. Pettitt Mutation Test

The Pettitt mutation test was used to divide the period of cotton production into smaller time periods that correspond to the five stages. The results are shown in Figure 2. This is a nonparametric test used to test mutation points and identify the mutation points of a sequence distribution and thereby determine the time of mutation [34]. With the help of the BreakPoints packages in R (v.4.1.0, R Core Team) [35], we used the Pettitt mutation test method to divide China's cotton production from 1949 to 2020 into five periods (i.e., 1949–1967, 1967–1981, 1981–1992, 1992–2006, and 2006–2020).

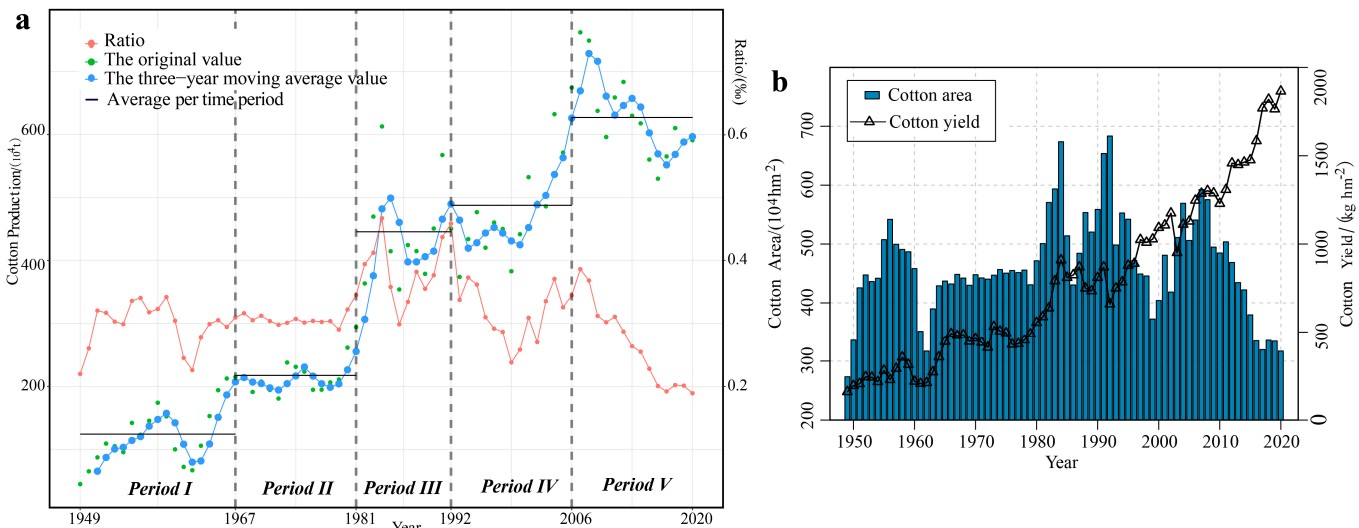

**Figure 2.** Inter−annual variations of cotton cultivation in China from 1949 to 2020: (**a**) cotton production ($10^4$ t) and ratio of cotton area to crop area, (**b**) cotton area ($10^4$ hm$^2$), and cotton yield (kg hm$^{-2}$).

### 2.3.3. Geographical Detector Model

The geographic detector model (GDM) is a widely used method for exploring the driving factors that influence spatial variation. This model may be used to explore the relationship between explanatory and dependent variables without any restrictions or assumptions in terms of variables. In addition, this method is immune to multiple collinearities [36–38]. The GDM consists of four parts: a factor detector, an interaction detector, an ecological detector, and a risk detector. In this study, the factor detection was used to explore the individual effect of impact factors on cotton yield, and the interaction detection was used to evaluate the joint effects of natural and agricultural technology factors on cotton cultivation when they interacted.

### 2.3.4. Factor Detector

The factor detector, characterized by the *q*-value, represents the explanatory capability of variable X for the dependent variable Y. The power of the influencing factors (*q*) on the distribution of cotton cultivation can be written as follows:

$$q = 1 - \sum_{i=1}^{L} n_i \partial_i^2 / n \partial^2,$$

where *i* is the number of stratifications (classes) of variable Y or factor X; $\partial_i^2$ and $\partial^2$ are the variance in the Y value of layer *i* and in the whole region, respectively; and *n* and $n_i$ are the number of units in the whole region and in layer *i*, respectively. *q* is the explanatory power of the driving force X to Y; the larger *q* is, the stronger the power of explanatory variables to explain related variables. The range of *q* is 0–1.

#### 2.3.5. Interaction Detector

The interaction detector is used to identify the interaction between natural and agricultural technological factors. Specifically, it evaluates the impact of factors $x_1$ and $x_2$ on the explanatory power of dependent variable cotton yield. It computes the interaction value $q(x_1 \cap x_2)$ for the two explanatory variables $x_1$ and $x_2$ and compares it with the original $q$ value of two explanatory powers to determine the type of interaction. There are three distinct interaction types as follows: superlinear interaction $[q(x_1 \cap x_2) > q(x_1) + q(x_2)]$, dual-factor interaction $[q(x_1 \cap x_2) > \text{Max}(q(x_1)\ q(x_2))]$, and uni-weaken interaction $[q(x_1 \cap x_2) < \text{Min}(q(x_1)\ q(x_2))]$.

We apply the GDM to analyze the influence of driving factors on cotton yield. Here, the cotton yield is considered the dependent variable, and the independent variables contain both anthropogenic and natural factors. It is important to note that the explanatory variables must be categorical variables. To address this, the explanatory variables are discretized by using the "optidisc" function via the GD package in R (v.4.1.0, R Core Team). The continuous independent variables are discretized into three to seven levels by using the optimal discretization method.

While applying the influencing factors analysis from 1980 to 2020, we removed rows with missing values. The data for analyzing how human–natural factors affect cotton cultivation consist of an original array of 680 records (i.e., 17 provinces × 40 years) and a validated array of 561 records (82.50%).

### 3. Results

#### 3.1. Inter-Annual Variation of Cotton Cultivation

China's cotton production showed an overall fluctuating upward trend from 1949 to 2020 (CV = 56.44%, $p < 0.001$). From 1977 to 1985, the production growth rate reached $453.02 \times 10^4$ t (10 a)$^{-1}$ ($p < 0.01$). The fastest yield growth rate was $458.49 \times 10^4$ t (10 a)$^{-1}$ from 2002 to 2008 ($p < 0.01$). The mean cotton production in these five periods was $124.38 \times 10^4$, $217.46 \times 10^4$, $445.65 \times 10^4$, $487.74 \times 10^4$, and $627.23 \times 10^4$ t, respectively (Figure 2a). In comparison, mean cotton production in period III increased by 104.94% compared with period II, and mean cotton production in period V increased by 403.30% compared with period I. Cotton area as a fraction of total sown area of crops in China from 1949 to 2020 is basically consistent with the variation in cotton area, with both increasing and then decreasing (Figure 2a,b). Specifically, the overall fluctuation of the cotton area is not significant (CV = 17.72%), with a peak of $683.5 \times 10^4$ hm$^2$ in 1992. The area variation in the period 1949–1992 is more moderate (CV = 17.57%, $p < 0.1$), with an average area of $470.71 \times 10^4$ hm$^2$. The area variation from 2007 to 2020 decreases significantly (CV = 22.11%, $p < 0.001$), with a mean area of $428.36 \times 10^4$ hm$^2$, a decrease of 9 percentage points compared with the previous period. According to the results of the five phases, the average cotton yield in the five periods is $436.09 \times 10^4$, $448.26 \times 10^4$, $550.42 \times 10^4$, $493.47 \times 10^4$, and $443.13 \times 10^4$ hm$^2$, respectively. Cotton area is at a maximum in period III, with an increase of 26.21% compared with period I. On the contrary, cotton area decreased in period V by 19.49% compared with period III. Cotton yield in China increased from 1949 to 2020 (CV = 59.59%, $p < 0.001$), with a stable variation in the early stages and a significant increase in the later stage (Figure 2b). The average cotton yield for each of the five periods is 279.41, 473.53, 774.04, 978.65, and 1459.85 kg hm$^{-2}$, respectively. Cotton yields increased by 422.47% in period V compared with period I.

Based on the coefficient of variation (CV) analysis, the CVs of cotton production for the five stages are 40.43%, 13.67%, 20.47%, 17.58%, and 10.51% (CV$_1$ > CV$_3$ > CV$_4$ > CV$_2$ > CV$_5$), respectively. The CVs of cotton yield are 16.11%, 3.80%, 13.99%, 15.55%, and 21.98% for the five phases (CV$_5$ > CV$_1$ > CV$_4$ > CV$_3$ > CV$_2$). The CVs of cotton yield for the five phases are 34.69%, 10.39%, 13.02%, 16.96%, and 14.94% (CV$_1$ > CV$_4$ > CV$_5$ > CV$_3$ > CV$_2$), respectively.

*3.2. Cotton Yield Variation*

Spatial coefficients of variation of cotton yields from time period I to V were observed as 0.30, 0.38, 0.30, 0.32, and 0.24, respectively. This implies a declining trend in cotton yield dispersion, indicating a gradual convergence of cotton yield among various regions. Specifically, in time period I, high-yield cotton regions were >480 kg/hm$^2$, mainly in Zhejiang and Shanghai. In time period II, high-yield regions were >700 kg/hm$^2$, mainly in Zhejiang, Jiangsu, and Shanghai in the eastern coastal region. In time period III, high-yield regions were >900 kg/hm$^2$, mainly in Zhejiang, Hubei, and Gansu, of which Gansu reached 1077.69 kg/hm$^2$. In time period IV, high-yield regions were >1300 kg/hm$^2$, in areas mainly including Xinjiang and Gansu, with 1399.48 kg/hm$^2$ and 1527.24 kg/hm$^2$, respectively. In time period V, high-yield regions were >1700 kg/hm$^2$, mainly including Gansu, Liaoning, and Xinjiang, with yields of 1718.06 kg/hm$^2$, 1725.15 kg/hm$^2$, and 1949.58 kg/hm$^2$, respectively (Figure S1).

From 1949 to 2020, the cotton yield across Chinese provinces witnessed an upward trend. The Yellow River Basin displayed a gradual increase in growth rates, while the Yangtze River Basin exhibited a fluctuating pattern of decrease and increase. Meanwhile, growth rates in the northwest showcased consistent improvements.

To elaborate further, in time periods I to II, six provinces (Yunnan, Shaanxi, Tianjin, Liaoning, Xinjiang, and Hebei) had cotton yield increases of between 10 and ~100 kg/hm$^2$. Five provinces (Beijing, Jiangxi, Shandong, Gansu, and Anhui) increased by between 100 and ~200 kg/hm$^2$. Five provinces (Zhejiang, Henan, Hunan, Sichuan, and Hubei) increased between 200 and ~300 kg/hm$^2$. In addition, the cotton yield in Shanxi decreased slightly, while Jiangsu and Shanghai had the largest increases of 386.46 kg/hm$^2$ and 403.80 kg/hm$^2$, respectively (Figure 3a). In time periods II to III, five provinces (Jiangsu, Zhejiang, Shaanxi, Henan, and Anhui) had cotton yield increases of between 100 and ~300 kg/hm$^2$, six provinces (Liaoning, Hubei, Sichuan, Hebei, Shanxi, and Hunan) had increases of between 300 and ~400 kg/hm$^2$, and four provinces (Beijing, Jiangxi, Xinjiang, and Shandong) had increases of between 400 and ~500 kg/hm$^2$. In addition, cotton yield decreased slightly in Yunnan and Shanghai, while Tianjin and Gansu had the largest increases of 511.20 kg/hm$^2$ and 654.18 kg/hm$^2$, respectively (Figure 3b). In time periods III to IV, three provinces (Guizhou, Hubei, and Shandong) had cotton yield increases of between 10 and ~100 kg/hm$^2$, seven provinces (Anhui, Hebei, Henan, Jiangsu, Zhejiang, Yunnan, and Shaanxi) had increases of between 100 and ~200 kg/hm$^2$, five provinces (Liaoning, Shanxi, Beijing, Guangxi, and Jiangxi) had increases of between 200 and ~300 kg/hm$^2$, and four provinces (Tianjin, Hunan, Shanghai, and Gansu) had increases of between 300 and ~450 kg/hm$^2$. In addition, cotton yield in Sichuan decreased by 99.21 kg/hm$^2$, while Xinjiang had the largest increase of 590.41 kg/hm$^2$ (Figure 3c). In time periods IV to V, there were two provinces (Hubei and Hunan) with cotton yield increases of between 20 and ~30 kg/hm$^2$, two provinces (Beijing and Gansu) with increases of between 100 and ~200 kg/hm$^2$, eight provinces (Sichuan, Jiangsu, Anhui, Shandong, Henan, Tianjin, Hebei, and Shanghai) with increases of between 200 and ~300 kg/hm$^2$, and three provinces (Zhejiang, Guizhou, and Shanxi) with cotton yield increases of between 300 and ~400 kg/hm$^2$. The increase of cotton yield was between 300 and ~400 kg/hm$^2$ in three provinces (Zhejiang, Guizhou, and Shanxi) and between 450 and ~600 kg/hm$^2$ in four provinces (Jiangxi, Guangxi, Xinjiang, and Shaanxi). In addition, Liaoning and Yunnan had the largest increases of 865.59 kg/hm$^2$ and 959.34 kg/hm$^2$, respectively (Figure 3d).

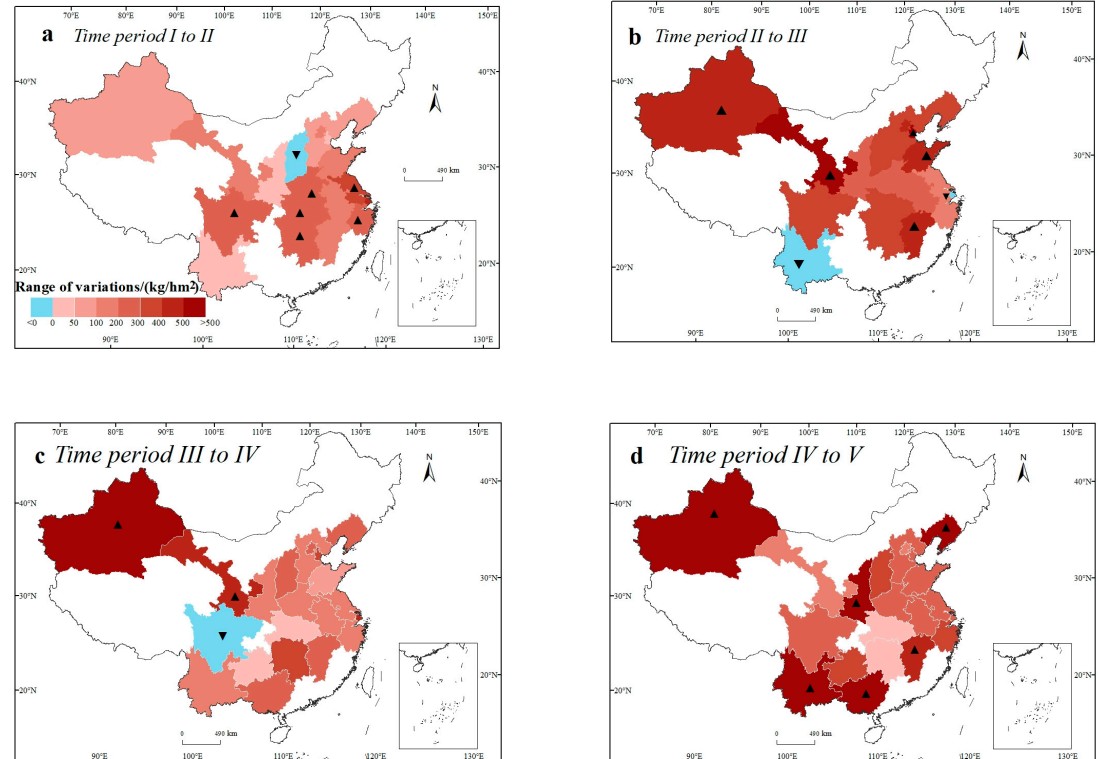

**Figure 3.** Maps for spatial distribution of cotton yield varied across China from 1949 to 2020. (**a**) Time periods I to II; (**b**) time periods II to III; (**c**) time periods III to IV; (**d**) time periods IV to V. Provinces with a triangle symbol represent wide range of variations, '▲' denotes positive change, and '▼' denotes negative change.

### 3.3. Analysis of Factors That Influence Cotton Yield in Different Regions

3.3.1. Factor Detection Analysis

Each individual anthropogenic and natural factor significantly influences cotton yield, with technological factors demonstrating the highest explanatory potency. Moreover, the interaction between technological factors and other variables yields considerable effects, all of which exhibit dual-factor enhancement or superlinear impacts.

In terms of temporal trends, climate factors remained relatively stable. However, significant variations were observed in technological, socioeconomic, and natural disaster factors (Figure S2). Bt transgenic insect-resistant cotton technology (Bt) adoption began in 1999, achieving widespread implementation in major cotton-planting provinces by 2001. Non-agricultural employment opportunities (O) and the cumulative power of agricultural machinery (M) showed a substantial increase in numerical values, with corresponding growth rates of 0.066/10 years ($R^2 = 0.87$) and 0.168/10 years ($R^2 = 0.95$), respectively. Meanwhile, natural disasters (D) demonstrated significant fluctuations, trending downwards with an index change rate of 1.531 ($R^2 = 0.42$). Table 2 shows the results for single-factor detection. Across China, all variables have successfully passed the 0.001 significance threshold, indicating that each variable significantly influences cotton yield. The explanatory power attributed to the cotton yield index drivers greater than 0.1 are, in descending order, F (0.503) > Bt (0.322) > M (0.248) > GEO (0.225) > EAT (0.196) > O (0.190) > PP (0.168) > TES (0.158) > ADT (0.143) > SSD (0.114) > D (0.080). This hierarchy signifies that technical indicators hold paramount importance as explanatory factors for China's cotton yield. Notably, the variables with the most substantial influence are fertilizer usage, Bt transgenic technology, and the total power of agricultural machinery. Subsequently, geographical positioning, non-agricultural employment prospects, and climatic indicators follow suit. Finally, the impact of disaster-affected area assumes the lowest priority.

**Table 2.** Correlating coefficients of single factors for cotton yield.

| Factor | China | | Yellow River Basin | | Yangtze River Basin | | Northwest Inland | |
|---|---|---|---|---|---|---|---|---|
| | *q* Value | Sig. | *q* Value | Sig. | *q* Value | Sig. | *q* Value | Sig. |
| EAT | 0.196 | <0.001 | 0.238 | <0.001 | 0.344 | <0.001 | 0.536 | <0.001 |
| PP | 0.168 | <0.001 | 0.069 | 0.182 | 0.062 | 0.013 | 0.148 | 0.015 |
| SSD | 0.114 | <0.001 | 0.114 | 0.010 | 0.062 | 0.107 | 0.193 | 0.019 |
| ADT | 0.143 | <0.001 | 0.100 | 0.003 | 0.269 | <0.001 | 0.460 | <0.001 |
| TES | 0.158 | <0.001 | 0.144 | 0.000 | 0.288 | <0.001 | 0.421 | <0.001 |
| TFB | 0.098 | <0.001 | 0.078 | 0.009 | 0.212 | <0.001 | 0.496 | <0.001 |
| MAT | 0.069 | <0.001 | 0.158 | <0.001 | 0.188 | <0.001 | 0.103 | 0.003 |
| D | 0.080 | <0.001 | 0.153 | <0.001 | 0.073 | 0.017 | 0.269 | 0.073 |
| O | 0.190 | <0.001 | 0.459 | <0.001 | 0.290 | <0.001 | 0.729 | <0.001 |
| M | 0.248 | <0.001 | 0.438 | <0.001 | 0.321 | <0.001 | 0.817 | <0.001 |
| F | 0.503 | <0.001 | 0.557 | <0.001 | 0.461 | <0.001 | 0.831 | <0.001 |
| Bt | 0.322 | <0.001 | 0.492 | <0.001 | 0.266 | <0.001 | 0.590 | <0.001 |
| GEO | 0.225 | <0.001 | - | - | - | - | - | - |

Both the Yellow River Basin region and the Yangtze River Basin region exhibit 11 factors that pass the 0.05 significance test, with 10 and 9 of these factors contributing >0.1 to the cotton area index, respectively. The highest *q* values, 0.557 for the Yellow River Basin region and 0.461 for the Yangtze River Basin region, are both associated with chemical fertilizer usage. In addition, all factors passed the 0.05 significance test in the northwest inland region, and the top three *q* values are for F (0.831), M (0.817), and O (0.729).

To delve into specifics, within the Yellow River Basin region, cotton yield is significantly influenced by factors such as fertilizer usage (0.557), transgenic technology (0.492), non-agricultural employment opportunities (0.459), and agricultural machinery power (0.438). The impact of maximum temperature (0.158) and disaster area (0.153) also remains pronounced ($p < 0.001$). Similarly, in the Yangtze River Basin region, cotton yield is notably affected by chemical fertilizer usage (0.461), agricultural machinery power (0.321), and effective accumulated temperature (0.344). Non-agricultural employment opportunities (0.290), average temperature (0.269), minimum temperature (TES = 0.288; TFB = 0.212), and maximum temperature (0.188) also exert significant influence on cotton yield in YARB. In Northwest China, cotton yield is heavily influenced by factors such as fertilizer usage (0.831), agricultural machinery power (0.817), and non-agricultural employment opportunities (0.729), surpassing the influence of other factors. Following these, transgenic technology and temperature factors play secondary roles. All these factors exhibit significant impact ($p < 0.001$). This reveals that rising temperatures and advancements in agricultural technology significantly contribute to increased cotton yield.

From a comparative perspective, it is evident that both agricultural technological advancements and climate factors significantly impact cotton yield, regardless of geographical or temporal scales. Furthermore, in the NC, fertilizer usage, agricultural machinery power, and non-agricultural employment opportunities possess a more profound influence on cotton production. Additionally, at a national level, temperature, precipitation, and solar radiation hold significant sway over cotton yield; however, within specific regions, cotton yield is primarily driven by changes in temperature.

### 3.3.2. Interaction Detection Analysis

Based on the interaction between the factors for cotton yield (Figure 4), chemical fertilizer use (F) and minimum temperature from *Flo* to *Bol* (TFB) have the highest explanatory power of 0.68 for the cotton yield of China, followed by the interactions of F and geographic region (GEO) with average daily temperature from *Sow* to *Squ* (ADT), which produce explanatory powers of 0.67 and 0.66, respectively. For cotton yield, the superlinear effect means that the synergetic effect of two factors exceeds the sum of their separate effects (e.g., F and MAT (0.65), M and TFB (0.53), and Bt and TFB (0.50)). The dual-factor effect means

that the synergetic effect of two factors is stronger than that of each factor alone (e.g., PP and F (0.63), Bt and PP (0.46), and GEO and SSD (0.27)) (see Figure 4a).

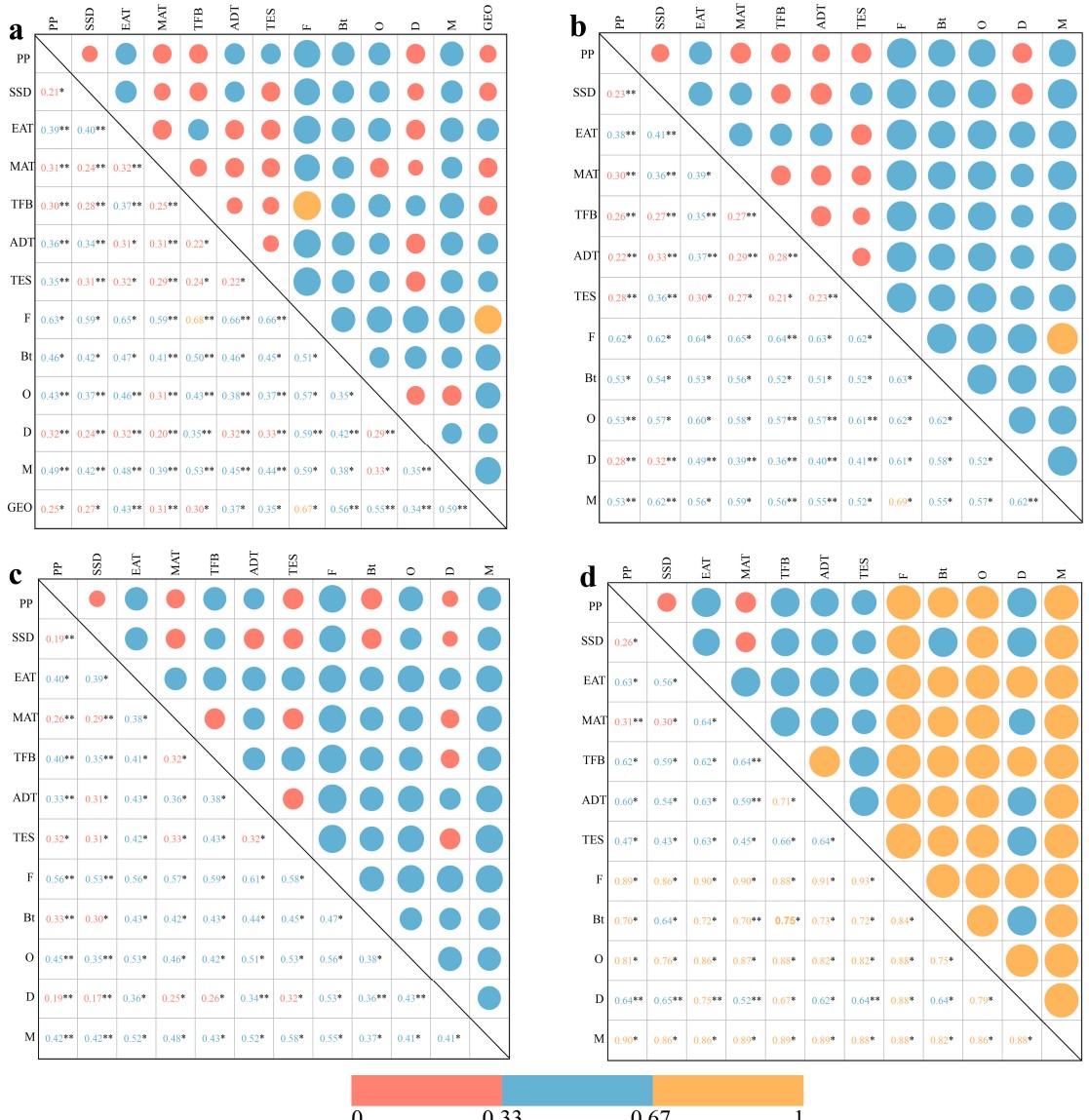

**Figure 4.** Influence due to interaction between CVs of cotton yield: (**a**) China's main cotton-producing provinces, (**b**) Yellow River Basin, (**c**) Yangtze River Basin, and (**d**) Northwest China. "**"": superlinear [$q(x_1 \cap x_2) > q(x_1) + q(x_2)$]; "*": dual-factor [$q(x_1 \cap x_2) > \text{Max}(q(x_1)\, q(x_2))$]. $q(x_1 \cap x_2)$ denotes the $q$ value for the interaction of factors $x_1$ and $x_2$ and $q(x_1)$ and $q(x_2)$ denote the separate effects of $x_1$ and $x_2$, respectively. As the $q$ value increases, the size of circles also increases.

Examining different regions within the YERB, fertilizer (F) and agricultural machinery power (M) exhibit the highest combined explanatory power of 0.69 for cotton yield. Following closely is the interaction between fertilizer and maximum temperature (MAT), which results in an explanatory power of 0.67. Both effects demonstrate superlinear relationships (Figure 4b). In contrast, the explanatory power of factor interactions for YARB ranges from 0.34 to 0.61, reflecting a relatively low impact of these factors on cotton yield (Figure 4c). Remarkably, in NC, the explanatory power of factor interactions ranges from 0.67 to 0.93, indicating a substantial contribution of technology and air temperature interactions to cotton yield (Figure 4d). Zooming in, whether at a national or regional scale, the interaction between fertilizer usage and various climate or technological factors plays a significant role in influencing cotton yield. Notably, interactions with minimum temperature yield the

most substantial impact. In a cross-regional comparison, NC displays the most pronounced influence on cotton yield. The interaction between agricultural technological advancement factors and social factors, coupled with other influencing factors, primarily drives changes in cotton yield in this area.

## 4. Discussion

### 4.1. Impacts of Climate Change on Crop Yield

Climate change has brought about variations in crop phenology and yield that differ by location, leading to accelerated shifts in crop spatial distribution [39]. Previous research has indicated a gradual transition of wheat cultivation in China towards high-yield regions like the Yellow and Huaihai Plain areas [40]. Additionally, due to increased heat-related damage to maize, production has gravitated from the Northeast Plain to the North China Plain and then back to the Northeast Plain over the past decade [41]. In the context of cotton yield, cumulative temperature and minimum temperature emerge as pivotal factors. While climate factors may not stand as the most dominant single factor, the interaction analysis reveals that the most influential combination of factors is the interplay between temperature and variables such as fertilizer usage (F), transgenic technology (Bt), and agricultural machinery power (M), both at national and regional scales. Through interregional comparisons, this study illustrates that the maximal explanatory power for cotton yield lies in single factors [$q$(F) = 0.831], and the interaction between factors [$q$(F∩TES) = 0.93], particularly pronounced in Northwest China. This underscores that a single factor, although exhibiting minor regional differences, can only account for results to a certain extent. When examined by region, the influence of climate, technology, and socioeconomic factors on YERB, YARB, and NC regions varies (Table 2). Notably, effective accumulated temperature (EAT) and maximum temperature (MAT) have the greatest impact on NC, followed by YERB. Non-farm employment opportunities (O), fertilizer usage (F), and total power of agricultural machinery (M) exhibit the highest influence on NC, with YARB ranking second. These results diverge from those of Han et al., who found that climate variation accounted for 54.42%, 58.10%, and 50% of cotton yield variability in YERB, YARB, and NC, respectively [42]. This discrepancy may be attributed to their emphasis on analyzing relative yield increase values, while our study primarily focuses on factors driving absolute yield changes. Research indicates that elevated temperatures lengthen the cotton growth period, facilitating increased yield by allowing for extended growth and development time [43]. However, if temperature rises excessively, the transpiration rate of crops accelerates, leading to heightened water loss and potential yield reduction [44]. Hence, in climatically sensitive regions for cotton cultivation, particularly major cotton-producing provinces like Xinjiang, the acceleration of agricultural water-saving technologies, effective fertilizer usage, transgenic technology adoption, and the cultivation of drought-resistant high-yield varieties should be considered [45,46]. This approach aims to curtail plant and soil root zone water loss, enhance water utilization efficiency, and ultimately ensure stable or elevated cotton yields.

### 4.2. Impact of Agricultural Technology on Spatial Pattern of Crop Cultivation

Agricultural technologies including insecticides, mulching, formula fertilization, intelligent drip irrigation, automated harvesting, and disease-resistant plant varieties can not only lower cost but also improve land productivity, capital utilization, and labor productivity. The application of agricultural technology depends on regional factors such as climate, economy, and population, whereas background natural resource such as topography and climatic conditions can limit the rate of technological innovation. Mismatching the two may prevent agricultural technological innovation from being effective and may exacerbate the waste of agricultural resources and increase costs, thereby countering agricultural productivity in the region and leading to new crop switching. For example, the rapid expansion of contiguous intensive crop cultivation correlates with the significant increase in mechanical equipment such as tractors and harvesters. Thus, the geographical bound-

ary of crop cultivation is mainly determined by human activity rather than by natural factors [47,48].

The results of this study show that technological factors are the most important factors for explaining the increase in cotton yield. The results of the single-factor analysis show that fertilizer use, Bt transgenic insect-resistant cotton, and total mechanical power are the three highest-ranked single factors at both national and regional scales (Table 2). In Northwest China, the explanatory power of the interaction between the three technological factors and other factors ranges from 0.7 to 0.93 (Figure 4d).

Technological innovation has driven the accelerated redistribution of cotton around the world. In the United States, the invention of the saw-tooth cotton gin and a variety of improvements have driven the cotton industry and elevated it to an impressive technological level [49]. In Mexico, Bt cotton is widely accepted by cotton producers and has proven to be efficient for pest control. The introduction of Bt cotton made it possible to reactivate this crop in 1996; in previous years, it was greatly reduced due to pests, production costs, and environmental concerns [50]. Drip irrigation under plastic mulching is widely used in northwest China and integrates plastic film mulching with surface drip irrigation. The water-soluble fertilizer is dissolved to form an aqueous solution. This strategy consumes 12% and 50% less water than conventional irrigation and sprinkler irrigation, respectively, and reduces the amount of fertilizer required by 15–20% [51]. Moreover, light and simplified nurseries and transplanting of cotton seedlings are new technologies destined to replace traditional nutrition-bowl nursery and transplanting [52]. The use of transplanting machines for cotton seedlings reduces labor intensity and improves efficiency. In addition, the cotton agribusiness in China has recently adopted plastic film mulching, reducing the effect of global warming on sowing time relative to crops such as wheat, maize, and potatoes [18].

### 4.3. Limitations of the Study and Future Prospects

Chinese cotton production from 1949 to 2020 was divided into five periods by applying the mutation test, and the data from each period were averaged. This treatment avoids the random error caused by natural disasters or financial crises that plague studies based on individual years. The years identified by the cotton production mutation test as important mutations are designated as key event milestones for cotton cultivation in China [33]. For example, in the 1960s, China issued a notice on good seed breeding, which motivated farmers to grow cotton. In the 1980s, the fraction of good seeds surpassed 80%, and the increase in the popularity of mulch led to a period of rapid development. China's acceptance into the World Trade Organization in the 2000s and the increase in its textile exports led to the rapid development of cotton production. Cotton production reached a record high of 7.60 million tons in 2007, accounting for about 30% of global cotton production. However, since 2007, cotton production has been declining. Affected by the financial crisis in 2008, cotton prices fluctuated widely. In addition, we study herein the spatial and temporal variations in cotton cultivation by dividing the period from 1949 to 2020 into five time periods. The results for cotton cultivation in China obtained herein are consistent with those of previous studies [3].

Due to limitations in data collection, this study does not consider how $CO_2$, mulch, or extreme weather affect cotton cultivation. However, these factors are still crucial. In addition to these aspects, another pivotal element of agricultural practices is soil characteristics [29]. Moreover, the GDM only detects the degree of impact of the impact factor, whereas the sign of the impact (positive or negative) must be determined by the underlying theory. Given the complex global changes of the future, safeguarding the stability of the cotton industry despite the threat of international trade conflicts, and guaranteeing the stable increase of cotton yields, require further study of the future spatial pattern of cotton cultivation, which can provide a scientific basis on which government can develop the appropriate strategies. Furthermore, it is possible to integrate satellite remote sensing

imagery, climate data, and soil information to develop models for predicting cotton yield across various regions in China.

## 5. Conclusions

A thorough investigation into the characteristics of variations in cotton cultivation and its driving mechanism in the main cotton-planting areas of China can provide valuable insights for decision making regarding climate change and crop management. In this study, we have examined the spatial and temporal patterns of cotton cultivation in China and established a five-level index system to assess factors influencing cotton yield. Employing the Pettitt mutation test and GIS spatial analysis, utilizing a three-year moving average, we have scrutinized the spatiotemporal dynamics of cotton cultivation in China from 1949 to 2020. Using the geographical detector model, we have quantitatively examined the influences of 13 chosen driving factors on changes in cotton yield across various regional scales between 1980 and 2020.

The main findings are as follows: all 17 Chinese provinces experienced advancements in cotton yield. The timeline from 1949 to 2020 was divided into five distinct periods, characterized by initially slow cotton yield growth, followed by a remarkable acceleration. This approach mitigated random errors stemming from individual year-based studies and designates them as pivotal milestones in Chinese cotton cultivation. Increases in factors such as minimum temperature, average temperature, effective accumulated temperature, and agricultural technologies like fertilizer usage, genetically modified varieties, and mechanized farming have contributed to the enhanced cotton yield. The importance of single factors influencing cotton yield of China in descending order was as follows: F > Bt > M > GEO > EAT > O > PP > TES > ADT > SSD > D. However, the effects of driving factors vary across regional scales. The 13 selected driving factors have collectively exerted amplified impacts on cotton yield variations due to interaction effects. The most significant interaction effects were observed between chemical fertilizer use and other driving factors. Specifically, the interaction between chemical fertilizer use (F) and minimum temperature (TES) has the greatest explanatory influence in Northwest China. Our finding emphasizes the key role of the interaction between agricultural technology and climate change in cotton yield variations, which should be meticulously considered in future research endeavors.

**Supplementary Materials:** The following supporting information can be downloaded at: https://www.mdpi.com/article/10.3390/agriculture13112132/s1.

**Author Contributions:** Conceptualization, Y.Z., W.J. and Y.Y.; Software, Y.Z. and B.Z.; Validation, B.Z.; Formal analysis, Y.Z. and B.Z.; Resources, Y.Z.; Writing—original draft, Y.Z.; Writing—review and editing, B.Z., Q.L., W.J. and Y.Y.; Supervision, Q.L.; Project administration, Q.L. and Y.Y.; Funding acquisition, Q.L. and Y.Y. All authors have read and agreed to the published version of the manuscript.

**Funding:** This research was funded by Project funded by the National Natural Science Foundation of China (No. 42271401), The Third Xinjiang Integrated Scientific Expedition Project (No. 2021xjkk02004), Special Funds for the Basic Research and Development Program in the Central Non-profit Research Institutes of China (No. 1610132023015, GJ2023-18-1), and the National Potato Industry Technology System (No. CARS-09).

**Data Availability Statement:** The data that support the findings of this study are available upon reasonable request from the authors.

**Conflicts of Interest:** The authors declare no conflict of interest.

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
