# Peer review of "Uncovering the Drivers and Regional Variability of Cotton Yield in China"

_agriculture, doi:10.3390/agriculture13112132_

Round 1

Reviewer 1 Report

Comments and Suggestions for Authors

1-In line 48, please include data on the quantity of production during the period from 1980 until the latest statistics.

2-On line 55, please include data on the amount of production in India

3-In line 54, please explain the advantages of genetically modified varieties over others

4-In line 97, I list the cotton regions with latitude and longitude as general regions

5-As for Figure A2, I suggest separating it into 5 separate maps, and explaining each stage in more detail

6-A review of the intervening factors requires more explanation and detail

7-Regarding the review of the five areas of increased productivity, please clarify the numbers that were and are now to clarify the differences, and I hope that they include a chronological sequence.

Author Response

The following are the point-to-point responses to the reviewers’ comments.

Reviewer #1:

  1. In line 48, please include data on the quantity of production during the period from 1980 until the latest statistics.

Response: Thank you for your suggestion. We have included the requested data and made significant revisions to the introduction section (see tracked version L50-56).

  1. On line 55, please include data on the amount of production in India

Response: Thank you for your revision suggestions. We have included the data and added references (see tracked version L50-56).

  1. In line 54, please explain the advantages of genetically modified varieties over others

Response: Thank you for your suggestions. We have provided a detailed explanation (see tracked version L61-66).

  1. In line 97, I list the cotton regions with latitude and longitude as general regions

Response: Thank you for your constructive comments and revision suggestions. We have added numerical range values for the longitude and latitude of the three regions, and also included latitude and longitude markings in Figure 1 (see tracked version L115-117).

  1. As for Figure A2, I suggest separating it into 5 separate maps, and explaining each stage in more detail

Response: Thank you for your comments. We have made textual revisions to this section and summarized the coefficient of variation in cotton production for the five stages, providing a detailed analysis of the fluctuations in each stage. We apologize and appreciate your feedback. It's important to note that this section primarily focuses on the temporal variations in cotton cultivation in China from 1949 to 2020, emphasizing continuity on the time scale. Therefore, separating the trend graphs for the five stages may not effectively convey the trends. Instead, we have provided a detailed explanation and textual summary for each stage based on our data analysis (see tracked version L268-301).

  1. A review of the intervening factors requires more explanation and detail

Response: Thank you for your constructive comments and revision suggestions. We have added supplementary explanations to the section on influencing factors (see tracked version L358-366).

In terms of temporal trends, climate factors remained relatively stable. However, significant variations were observed in technological, socioeconomic, and natural disaster factors (Fig. S2). Bt transgenic insect resistant cotton technology (Bt) adoption began in 1999, achieving widespread implementation in major cotton-planting provinces by 2001. Non-agricultural employment opportunities (O) and the cumulative power of agricultural machinery (M) showed a substantial increase in numerical values, with corresponding growth rates of 0.066/10 years (R2=0.87) and 0.168/10 years (R2=0.95), respectively. Meanwhile, natural disasters (D) demonstrated significant fluctuations, trending downwards with an index change rate of 1.531 (R2=0.42).

  1. Regarding the review of the five areas of increased productivity, please clarify the numbers that were and are now to clarify the differences, and I hope that they include a chronological sequence.

Response: Thank you for your constructive comments and revision suggestions. We have conducted a reanalysis of the influencing factors. The analysis results primarily include two time series trend graphs for the various influencing factors of Chinese cotton yield from 1980 to 2020. We have also added trend line analysis for factors showing significant changes. The main results have been included in the main text, while the graphs have been added to the supplementary materials (see tracked version L358-366).

Reviewer 2 Report

Comments and Suggestions for Authors

Reviewer Comments

The article requires these suggestions to meet the publication standards of Agriculture Journal, and my observations are as follows:

Firstly, the title needs improvement to better represent the study's purpose.

The introduction section appears to be somewhat weak and could benefit from further elaboration. It is essential to highlight the importance of the subject area and provide a clear understanding of the problem. Additionally, including a theoretical framework for the study is recommended. Moreover, I suggest adding more relevant literature and citations to strengthen the introduction.

The selection of the study area lacks a clear rationale, and this should be addressed. Furthermore, the manuscript lacks information about the sample size from 1949 to 2020, and selection criteria. These critical details should be included.

The conclusion section could benefit from the addition of information regarding limitations and suggestions for future research.

In general, I found the manuscript interesting, but it seems somewhat simplistic in its current form. Authors should incorporate suggested inputs and include more recent citations to enhance the paper's quality.

Comments on the Quality of English Language

Moderate editing of the English language required

Reviewer 3 Report

Comments and Suggestions for Authors

The manuscript entitled “Uncovering the primary drivers of regional variability in natural anthropogenic impacts on cotton yields in Chiba” presents a discussion on sustainability of cotton production in China using simple statistical approaches. The structure is rather simple and the methodology used for the research is not new. I do not see any novelty in the combination of climate variables and statistical variables for understanding the yield variability. However, the discussion on the drivers of the yield is really of important, as seen in the following publications in 2023. Please extend the discussion with their findings. Is there any common yield drivers? In China is cotton rainfed crop or not. While discussing, please consider similar constraints as well. 

*In the abstract, there are abbreviations, please make them clear. There are not well known abbreviations. Similarly, in the text once you define the abbreviations, there is no need to repeat them again and again in the text and please clarify the abbreviations in their first usage. 

*Would be more meaningful to make the training with %80 of the data.     

*In Section 3.2, what is high yield, what is low yield? For example in figure 3, all the plots have different color scales, making the interpretation not meaningful. Please clarify your point. 

*Would be interesting to apply SHAP as well to see the drivers of the yield and compare the results with your statistical findings.

Climate variation explains more than half of cotton yield variability in Chinahttps://doi.org/10.1016/j.indcrop.2022.115905

Explainable Artificial Intelligence for Cotton Yield Prediction With Multisource Data, doi: 10.1109/LGRS.2023.3303643

Comments on the Quality of English Language

Editing is required specifically for abbreviations and etc.

Round 2

Reviewer 2 Report

Comments and Suggestions for Authors

Good effort to incorporates the all suggestions.